# Molecular Typing Reveals Distinct *Mycoplasma genitalium* Transmission Networks among a Cohort of Men Who Have Sex with Men and a Cohort of Women in France

**DOI:** 10.3390/microorganisms10081587

**Published:** 2022-08-06

**Authors:** Jennifer Guiraud, Marion Helary, Chloé Le Roy, Eric Elguero, Sabine Pereyre, Cécile Bébéar

**Affiliations:** 1ARMYNE Team, UMR 5234, Microbiologie Fondamentale et Pathogénicité (MFP), University of Bordeaux, Centre National de la Recherche Scientifique (CNRS), 33000 Bordeaux, France; 2Bacteriology Department, French National Reference Centre for Bacterial Sexually Transmitted Infections, Bordeaux University Hospital, 33000 Bordeaux, France; 3Maladies Infectieuses et Vecteurs: Écologie, Génétique, Évolution et Contrôle (MiVEGEC), Institut de Recherche pour le Developpement (IRD), University of Montpellier, Centre National de la Recherche Scientifique (CNRS), 34000 Montpellier, France

**Keywords:** *Mycoplasma genitalium*, antimicrobial resistance, sexually transmitted infection, macrolides, fluoroquinolones, *mgpB*, MG309, typing

## Abstract

*Mycoplasma genitalium* causes sexually transmitted infecti.ons in men and women. Treatment failures to macrolides and fluoroquinolones have been reported worldwide. Although the *mgpB* typing method has often been used in *M. genitalium*-infected men who have sex with men (MSM), limited typing data are available for *M. genitalium*-infected women. In this study, we aimed to investigate the genetic relationship between *M. genitalium* strains and their antibiotic resistance profile in a cohort of MSM (86.2% on HIV preexposure prophylaxis [PrEP], 13.8% HIV positive) and a large cohort of women using *mgpB*/MG309 typing. The *mgpB* types were determined in 374 samples from 305 women and 65 MSM. Three MSM and one woman had two concurrent or subsequent samples. Macrolide and fluoroquinolone resistance-associated mutations were searched in the 23S rRNA as well as *parC* and *gyrA* genes. The *mgpB* phylogenetic construction revealed three large clusters that differed according to sexual practices and geographical origin of patients. The prevalence of macrolide and fluoroquinolone resistance was significantly higher in MSM compared with women (95.4% vs. 14.1% and 30.6% vs. 7.2%, *p* < 0.001, respectively). The macrolide resistance spread was polyclonal in both populations, but clonal diffusion of two dual-resistant types was observed in PrEP users in association with high antibiotic pressure and dense connectivity in this population.

## 1. Introduction

*Mycoplasma genitalium* is a fastidious sexually transmitted pathogen. *M. genitalium* is responsible for urethritis and proctitis in men and is associated with cervicitis, pelvic inflammatory diseases, and infertility in women [1]. An increasing occurrence of treatment failures to first- (macrolides) and second-line (fluoroquinolones) treatments has been reported worldwide [1,2]. Acquired mutations in the 23S rRNA gene (positions 2058 and 2059, *Escherichia coli* numbering) are associated with macrolide resistance, whereas acquired mutations in the quinolone resistance determining regions (QRDR) of ParC (G81C, S83R, S83I, D87N, and D87Y) lead to fluoroquinolone resistance in *M. genitalium* [1,2,3,4]. The French National Reference Center (NRC) for bacterial sexually transmitted infections (STIs) reported rates of macrolide and fluoroquinolone resistance of 42.1% and 15.8% in 2020 in France, respectively [5]. Antibiotic resistance transmission is thus concerning even though *M. genitalium* strains infecting heterosexual men and women are less likely to be resistant than those isolated from men who have sex with men (MSM) and MSM taking HIV preexposure prophylaxis (PrEP) [2,6,7,8]. In this context, a better understanding of the determinants of *M. genitalium* transmission in sexual networks and of the association between genotypes and antimicrobial resistance is necessary.

Molecular typing methods are commonly based on the analysis of the *mgpB* gene, which encodes the hypervariable adhesin MgPa [9]. The combined analysis of the *mgpB* sequence polymorphism and the number of repeats in the MG309 locus, encoding a putative lipoprotein, allowed us to distinguish persistent and recurrent infections among follow-up samples [7,10,11,12,13]. The molecular epidemiology of *M. genitalium* infections using the *mgpB* or *mgpB*/MG309 typing method was also often used to elucidate transmission networks among MSM [7,10,11,13,14,15]. Nevertheless, limited typing data are available for *M. genitalium* strains infecting women, as epidemiological studies on women are commonly based on small sample sizes [11,13,15,16,17,18].

In this study, we explored the genetic relationships based on *mgpB/MG309* typing among *M. genitalium*-positive specimens collected from two distinct populations: a large cohort of women and a cohort of MSM, most of whom were on PrEP. We also evaluated the prevalence of macrolide and fluoroquinolone resistance in both cohorts to investigate the link between the *mgpB* sequence types (ST) of *M. genitalium* strains and their antimicrobial resistance (AMR) profile. Two analyses of the *mgpB* types distribution were conducted, one on the overall population and one on each cohort separately. 

## 2. Materials and Methods

### 2.1. Sample Collection

In this retrospective study, we used remnants of DNA extracts stored at −80 °C and obtained from samples collected between January 2018 and December 2019 at the French NRC for bacterial STI (STI NRC) in the Bacteriology Department of the Bordeaux University Hospital (Bordeaux, France). Samples from two distinct populations, a population of *M. genitalium*-positive MSM on PrEP or HIV-positive and a population of *M. genitalium*-positive women, were collected. Samples from the MSM cohort were collected from patients visiting the Infectious and Tropical Diseases Department of the Bordeaux University Hospital or STI clinics located in metropolitan France. Of note, at the time of the study, the screening for *M. genitalium* in France was based on the 2016 European guideline on *M. genitalium* infection, which recommended testing in individuals with a high-risk sexual behaviour [19]. Screening practices changed later according to the publication of the BASHH guideline for the management of *M. genitalium* infection in 2018, and the update of the European guideline in 2021 [20,21]. Samples from the women cohort were collected from women included as part of annual prevalence surveys carried out by the STI NRC in metropolitan and overseas France (La Reunion Island in the Indian Ocean region, French Guyana on the northeast coast of South America, and French Polynesia in the South Pacific region). Samples were systematically collected if a successful *M. genitalium* macrolide resistance status had been obtained using the ResistancePlus kit (SpeeDx) [22].

The demographical and epidemiological characteristics of individuals, including sex, age, sampling site, and *C. trachomatis* coinfection, were anonymously collated for both cohorts. For MSM, clinical symptoms, macrolide treatment anteriority, and *N. gonorrhoeae* coinfections were also collected.

### 2.2. Laboratory Procedures

A single nucleotide polymorphism (SNP) analysis of the adhesin gene *mgpB* and a short tandem repeat analysis of the putative lipoprotein gene MG309 were performed. *mgpB* types were determined after amplification and sequencing of a 281-bp region in the *mgpB* gene [9]. A new real-time PCR approach using the MgPa-1/3 primer set and the MgPa-380 probe was developed from a previous end-point PCR [23]. A volume of 5 µL of DNA extract was added to 20 µL of the reaction mixture (2X Light Cycler^®^480 Probes Master Mix (Roche) and 0.3 µM of each primer and probe). PCR was performed on a Light Cycler^®^ 480 instrument (Roche Diagnostic, Indianapolis, IN, USA). Cycling consisted of an activation cycle of 95 °C for 10 min followed by 45 cycles of 95 °C for 15 s and 60 °C for 1 min. PCR products were sent to Eurofins Genomics (Germany) for Sanger sequencing. The sequencing data were analysed using BioEdit 7.2.5 software (Isis Pharmaceuticals, Inc., Carlsbad, CA, USA). The obtained *mgpB* sequences were compared with the reference *M. genitalium* G37 strain sequence (GenBank accession no. NC_000908.2) and the 256 *mgpB* sequences published to date (Appendix A), according to Dumke et al. numbering [11]. MG309 types were determined by amplification and sequencing of a 581-bp region in the MG309 locus, as previously described [7,24]. Mutations in the QRDRs of the *parC* and *gyrA* genes were determined as previously described [4,25].

### 2.3. Data Analysis

To evaluate the genetic variability, allelic frequencies and the genetic diversity per locus and population were calculated using FSTAT software (version 2.9.3, Lausanne, Switzerland). The Hunter–Gaston discriminatory index was also calculated for unrelated samples [26]. To visualize the genetic relationships between strains, a phylogenetic tree based on *mgpB* typing was constructed using the maximum likelihood clustering method. MEGA software (version 7.0, University Park, PA, USA) was used for tree elaboration, and iTOL software (version 6.0, Heidelberg, Germany) was used for tree visualization and annotation. Statistical analysis was performed using online conventional statistical tests (http://biostatgv.sentiweb.fr, accessed on 2 August 2022). A comparison of age averages was performed using the Wilcoxon signed-rank test. The distribution of categorical variables was compared using chi-square or Fischer’s exact tests as appropriate. A *p* value of <0.05 was considered statistically significant.

## 3. Results

### 3.1. Sample and Patient Characteristics

A total of 632 *M. genitalium*-positive samples with an available macrolide resistance status were selected for this study. For these 632 samples, successful *mgpB* typing was obtained for 374 (59.2%) samples collected from 370 patients (Table 1).

A total of 306 samples from 305 *M. genitalium*-positive women were selected as the women cohort and consisted of 259 vaginal swabs (84.6%), 44 urine samples (14.4%), and three rectal swabs (1%). A total of 68 samples from 65 *M. genitalium*-positive MSM (86.2% on PrEP, 13.8% HIV positive) were selected as the MSM cohort and consisted of 38 rectal swabs (55.9%), 29 urines (42.7%), and one pharyngeal swab.

The epidemiological characteristics of the patients showed that the MSM were significantly older than the women (*p* < 0.001) (Table 1). The geographical origins of individuals did not differ significantly between the cohorts in metropolitan France (*p* = 0.286). In overseas France, only *M. genitalium*-positive women were recruited (57%), most of them were living on Reunion Island (83.9%; 146/174) followed by Guyana (10.3%; 18/174) and French Polynesia (5.7%; 10/174). Regarding coinfections, there was no significant difference regarding *C. trachomatis* coinfection between the cohorts (*p* = 0.084). *N. gonorrhoeae* coinfection was reported in 14 of 60 MSM (23.3%), and symptoms were reported in 43 of 57 of MSM (75.4%, 39 on PrEP, 4 HIV positive).

### 3.2. Antimicrobial Resistance

The macrolide resistance prevalence was 95.4% (62/65) among MSM and was only 14.1% (43/305) among women (*p* < 0.001) (Table 1). In addition, 89.7% (35/39) of MSM had received a previous macrolide treatment. Regarding fluoroquinolone resistance, sequencing of the *parC* and *gyrA* QRDR was achieved for 342 and 286 samples from 338 and 283 patients, respectively (Table 2).

The most prevalent ParC mutation was Ser83Ile (68.1%; 32/47). Four distinct GyrA mutations were detected in four different patients (1.4%). Gly93Cys and Met95Ile were detected in association with the Ser83Ile ParC mutation, whereas the Ala105Thr and Asp107Asn mutations were detected alone.

Considering mutations likely to have clinical significance (Gly81Cys, Ser83Arg, Ser83Ile, Asp87Asn, and Asp87Tyr) [3,4,14], the prevalence of fluoroquinolone resistance was significantly higher in MSM than in women (30.6% (19/62) vs. 7.2% (20/276); *p* < 0.001). The dual macrolide and fluoroquinolone resistance rate was also significantly higher in MSM than in women (30.6% (19/62) vs. 3.3% (9/276); *p* < 0.001).

### 3.3. Genetic Diversity and Population Structure

Among the 374 specimens analysed, 110 different *mgpB* types were identified with a *mgpB* gene diversity of 0.938. We describe here 67 new *mgpB* types numbered ST257 to ST323 (accession nos. MZ553845-MZ553914) compared to the 256 *M. genitalium* types previously reported to date and summarized in Appendix A. The most prevalent *mgpB* types were ST2 (14.4%; 54/374), ST7 (13.4%; 50/374), and ST4 (12.8%; 48/374). The MG309 type was obtained for 298 samples (79.7%). A total of 13 different types were identified with a gene diversity of 0.863. The *mgpB*/MG309 typing method identified 157 different genotypes, resulting in a Hunter–Gaston discriminatory index of 0.989, after having removed the one subsequent and the three concurrent samples. For two patients (nos. 1 and 2; Figure 1), the same genotype in each individual was detected 90 and 70 days apart, respectively. The strains were macrolide and fluoroquinolone resistant with no evolution of the AMR profile, suggesting the persistence of the same dual-resistant *M. genitalium* strain. For one patient (patient 3), a different genotype was identified after 35 days, suggesting a new infection. For two patients (nos. 2 and 4), the same genotype was detected in genital and rectal specimens, suggesting infection by the same strain at different anatomical sites.

The genetic distribution of the 374 *M. Genitalium* strains displayed by a maximum likelihood tree (MLT) based on the *mgpB* gene polymorphism (Figure 1) revealed three large clusters (Clusters A, B, and C). These clusters differed according to the epidemiological characteristics of the patients and the AMR profile of the strains (Table 3). The proportion of MSM (27.7%) and the proportion of rectal swabs (18.9%) were significantly higher in cluster C than in the two other clusters. Indeed, 60.0% (39/65) of MSM belonged to Cluster C, while Clusters A and B were mostly composed of women (88.6% and 88.7%, respectively). Despite numerous similarities found between Clusters A and B, individuals belonging to Cluster A mainly lived in French overseas territories (55.0%), whereas patients in Cluster B mainly lived in metropolitan France (63.8%; *p* = 0.007). Regarding antibiotic resistance, 37.8% of *M. genitalium* strains were resistant to macrolides in Cluster C, which was significantly higher than that noted in Cluster A (20%; *p* < 0.001). Interestingly, the fluoroquinolone resistance rate was significantly lower in Cluster C than in Cluster B (8.3% vs. 17.8%; *p* = 0.041).

In a second analysis, we focused on the most prevalent *mgpB* types to explore the correlation with epidemiological characteristics of patients and the occurrence of resistance-associated mutations. Although *mgpB* gene diversity was not significantly lower among MSM than in women (0.879 vs. 0.937; *p* = 0.121), a distinct *mgpB* type distribution was observed between both populations. ST7 was significantly more prevalent among women than among MSM (15% vs. 5.9%; *p* = 0.049). In contrast, ST4, 159, 108, and 113 were significantly more prevalent among MSM than among women (27.9% vs. 9.5%, *p* < 0.001; 16.2% vs. 0.3%, *p* < 0.001; 8.8% vs. 0.3%, *p* < 0.001, and 8.8% vs. 1.6%, *p* = 0.006, respectively). No difference in the distribution of *mgpB* types was found between MSM on PrEP and HIV-positive MSM. Of note, *mgpB* gene diversity was lower with a borderline statistical significance within rectal swabs than in other sampling sites (0.855 vs. 0.941; *p* = 0.052) with 56.1% of rectal samples belonging to ST4, 113, and 108. In contrast, ST159 was significantly more often detected in urine than in other sampling sites (11% vs. 1.3%, *p* < 0.001).

ST4, 159, and 108 *M. genitalium* strains were significantly more prevalent among macrolide-resistant than among susceptible strains (23.6% vs. 8.5%, *p* < 0.001; 10.6% vs. 0.2%, *p* < 0.001, and 5.6% vs. 0.4%, *p* = 0.003, respectively). ST108 and ST159 *M. genitalium* strains were significantly more prevalent among fluoroquinolone-resistant than among susceptible strains (14.6% vs. 0.3%; 26.8% vs. 0%, respectively, *p* < 0.001). Despite a distinct distribution of *mgpB* types according to the AMR profile, *mgpB* gene diversity was not significantly lower among macrolide- and/or fluoroquinolone-resistant strains than among susceptible strains (0.905 vs. 0.938 and 0.885 vs. 0.936; *p* = 0.268 and *p* = 0.186, respectively).

In a third analysis, we focused on the *mgpB* distribution within each cohort. Focusing on the *mgpB* distribution within the women cohort, ST7 was significantly more prevalent among women living in metropolitan France than among women living in overseas France (20.5% vs. 10.9%, *p* = 0.024), whereas ST257 was only detected in women living in overseas France (*p* < 0.001). ST8 was more often detected in specimens coinfected with *C. trachomatis* than in those infected with *M. genitalium* only (10% vs. 1.9%, *p* = 0.030). In the female cohort, no significant difference in *mgpB* type distribution was observed according to the sampling site or the macrolide and fluoroquinolone resistance status.

Concerning the *mgpB* distribution in the MSM cohort, we noted that ST159 was significantly more often detected in urine than in rectal swabs (27.6% vs. 7.9%, *p* = 0.046), whereas the prevalence of ST4, 113, and 108 was not significantly different in rectal swabs and urine (36.8% vs. 17.2%, 10.5% vs. 6.8%, and 13.2% vs. 3.4%, *p* = 0.10, *p* = 0.69, and *p* = 0.22, respectively). All *M. genitalium* strains belonging to ST4 were susceptible to fluoroquinolones.

Additionally, 10 MSM on PrEP (17.9%), of whom 8 lived in Paris (80%), were infected by a dual-resistant ST159 strain harbouring a 23S rRNA and the Ser83Ile ParC mutations and presenting close MG309 patterns ranging from 9 to 11 repeats. Similarly, six (10.7%) PrEP users living either in Paris or elsewhere in France were infected by a *M. genitalium* strain belonging to ST108. All six *M. genitalium* strains were dual resistant, harbouring 23S rRNA and the Ser83Ile ParC mutations and close MG309 patterns (9 or 10 repeats).

## 4. Discussion

Although the *mgpB* typing method has often been used to elucidate transmission networks among MSM [7,10,11,13,14,15], we reported here the first typing study focusing on a large cohort of women with an evaluation of macrolide and fluoroquinolone resistance. Additionally, we provide a comprehensive table of previously reported *M. genitalium* types along with the 67 new *mgpB* types reported in this study (Appendix A).

Using *mgpB*/MG309 typing, we revealed three large clusters that differed according to gender, male sexual practices, and the geographical origin of patients. Indeed, most MSM (60%) and rectal swabs (66%) belonged to Cluster C, whereas most women (66.6%) and vaginal swabs (67.9%) belonged to Clusters A and B. Thus, despite a high genetic heterogeneity in *M. genitalium* [7,11,13,27], *mgpB*/MG309 typing reveals distinct transmission dynamics related to distinct sexual networks in women and MSM. However, the most prevalent *mgpB* types, ST2, ST7, and ST4, were present in both populations, suggesting connections between MSM and women, as previously reported [11,15]. Additionally, we observed in this study that Cluster B was mostly composed of women living in metropolitan France (63.8%), whereas Cluster A was mostly composed of women living in overseas France (55%). ST257 was only detected in samples from women living on La Reunion Island. Thus, our data suggest limited connections between sexual networks of women from metropolitan and overseas France, which may be explained by the long distance between the French overseas territories and metropolitan France (approximately 10,000 km).

Notably, sampling sites were strongly associated with sex in our study, as only three rectal swabs were collected from women. Although a lower *mgpB* gene diversity was reported in rectal samples, there was no association between *mgpB* types and sampling sites within each cohort, except for ST159, which was more often detected in urine in the MSM in the PrEP cohort. Nevertheless, further investigations using a larger sample size are needed to confirm the predominance of *mgpB* type 159 in the urine of MSM and to determine whether this ST might be more easily transmitted by genital intercourse than by rectal intercourse. 

Higher AMR rates were unsurprisingly found among MSM than among women [2,8]. Besides MSM status, the difference in macrolide resistance between both cohorts may also be driven by lower consumption of antimicrobials in offshore populations. Although the French guidelines for the management of *M. genitalium* infections are the same in metropolitan and overseas France [19,21], lower accessibility to STI treatment of populations living in remote areas can occur. Nevertheless, the macrolide resistance rate among MSM (95.4%) was greater than that in previous studies conducted among MSM in Europe and Australia, in which rates ranged between 57% and 84% [6,7,11,14,15]. A total of 30.6% of MSM harboured an *M. genitalium* strain with dual-class resistance, which is twofold higher than that previously reported in MSM in Germany and Australia [11,14]. In our population, although the previous antimicrobial usage data were unavailable for the women cohort, 89.7% of MSM had previous macrolide treatment, revealing a strong antibiotic selection pressure. Indeed, multidrug- or extensively drug-resistant *M. genitalium* strains have been more often found among samples from MSM with a high-risk sexual behaviour, such as PrEP users and HIV-positive patients [7,8,11,28]. In our study, the most prevalent *mgpB* types among MSM, ST159, ST108, and ST4, were also the most resistant strains. Our analysis showed that AMR was not associated with the *mgpB* type but rather with the MSM status. Regarding ST4, several studies reported its predominance among MSM in Europe [7,10,11,13,15] and its possible association with macrolide resistance [10,11,13] and with easier transmission by anal intercourse [7]. In our study, ST4 was predominant among MSM; nevertheless, ST4 was not significantly associated with rectal samples (*p* = 0.103) or with macrolide resistance (*p* = 0.554). Of note, in the ST4 *M. genitalium* strains isolated from women, only 25.0% (7/28) carried macrolide resistance-associated mutations. Moreover, the multivariate analysis did not reveal any significant association of the most prevalent *mgpB* types with sex and macrolide resistance status, due to the collinearity between these related variables (Appendix A). 

Surprisingly, fluoroquinolone resistance was higher in Cluster B, comprising 88.7% of women predominantly from metropolitan France, than in Cluster C, comprising the highest proportion of MSM. The occurrence of fluoroquinolone resistance-associated mutations in *M. genitalium* is probably not only related to STI management. Indeed, nonadherence to the updated therapy guidelines for uncomplicated urinary tract infections has been reported in Europe [29], suggesting the possibility of inappropriate prescription of fluoroquinolones in women.

Overall, no association was noted between *mgpB* types and AMR in this study, providing evidence that AMR spread is polyclonal among men and women [7,13,14,15,18,27,30]. Macrolide resistance-associated mutations appear to occur in independent *M. genitalium* strains under antibiotic selection pressure [8,15]. However, in the MSM cohort, dual-resistant ST159 and ST108 *M. genitalium* strains, each harbouring a close number of repeats in MG309 and an identical Ser83Ile fluoroquinolone resistance-associated mutation (Appendix A), suggest the diffusion of two clones, consisting of MSM on PrEP. Thus, clonal diffusion of MDR *M. genitalium* strains may have occurred in the MSM community in link with the dense connectivity of sexual networks.

In addition to the lack of information regarding the sexual practice of women, a limitation to conduct this study was the poor sensitivity (374/632, 59.2%) of the *mgpB* typing method applied directly to clinical samples despite the development of a real-time PCR-based method. In this study, the poor sensitivity was related to DNA degradation because of the long period between sampling and PCR testing, leading to uninterpretable results after the Sanger sequencing. Additionally, among the 493 successful *mpgB* amplifications initially obtained, 9.5% (47/493) had an uninterpretable result after the Sanger sequencing due to a mixture of two or more chromatogram peaks. Indeed, sequencing troubleshooting was previously reported by several authors [24,31,32], whereby two recent studies using amplicon deep sequencing demonstrated the existence of mixed SNP types in the 23S rRNA, *parC*, *gyrA*, and *mgpB* genes [31,32]. These observations suggest the potential existence of infections by several *M. genitalium* strains or the presence of type variants in a single patient. Given the current typing difficulties, the use of next-generation sequencing methods based on whole-genome sequencing directly in clinical specimens may provide higher resolution to investigate *M. genitalium* strain relationships and dynamics.

## 5. Conclusions

Here, we reported the occurrence of distinct *M. genitalium* clusters depending on sex, geographical origin of infected patients, and sexual practices in men. In *M. genitalium*, the macrolide resistance spread was polyclonal in women and MSM. However, using the *mgpB* typing method, clonal diffusion of two specific *mgpB* types associated with dual resistance was suggested in small groups of PrEP users in metropolitan France. The spread of dual resistant *M. genitalium* strains in high-risk groups related to high antibiotic pressure and dense connectivity is concerning. The surveillance of AMR in these populations remains necessary.

## Figures and Tables

**Figure 1 microorganisms-10-01587-f001:**
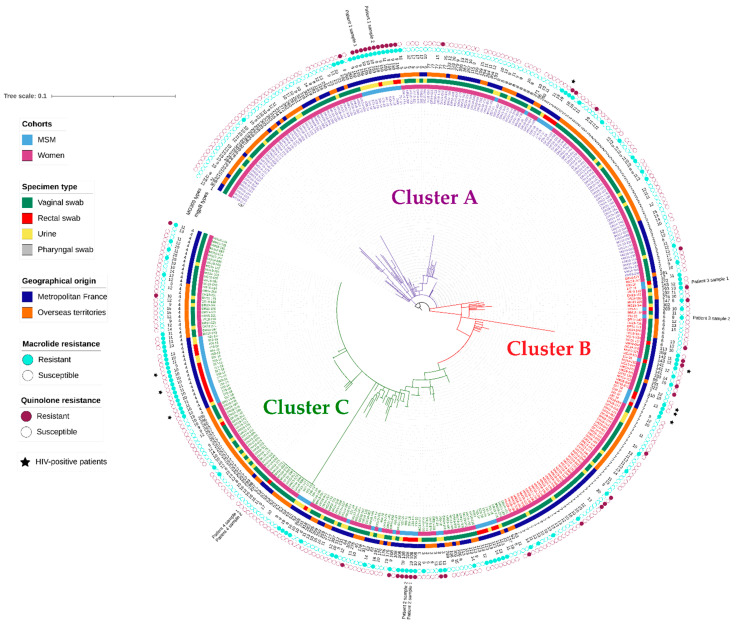
Maximum likelihood tree based on the *mgpB* type of the 374 *M. genitalium* strains collected from 305 women and 65 MSM. The tree was constructed using a T92 G+I model. Branch support values were generated from 1000 bootstrap replicates. The sequence of the *M. genitalium* G37 strain was used as a reference. The phylogenetic tree was annotated with (from the centre to the outside) the strain name, the cohort, the individual geographical origin, the sampling site, the *mgpB* sequence type (from 1 to 323), the MG309-STR type of the strain (from 8 to 20, if available), and the macrolide and fluoroquinolone resistance profile of the strain (if available). Clusters A, B, and C are represented by branches and strain names in purple, red, and green, respectively. Concurrent or subsequent samples from four patients are highlighted. HIV-positive patients are symbolized by a black star. For better reading, a clearer figure is available as Appendix A.

**Table 1 microorganisms-10-01587-t001:** Characteristics of the 370 *M. genitalium*-positive patients with successful *mgpB* typing results.

Characteristics	Women *n* = 305	MSM ^a^ *n* = 65	*p* Value
**Age**			
Mean +/− SD	26.3 +/− 7.9	36.6 +/− 12.5	<0.001 ^b^
Median (IQR)	24 (14–52)	33 (20–75)	
**Geographical origin—no. (%)**			
Metropolitan France	131 (43.0%)	65 (100.0%)	
Paris region	82 (62.6%)	37 (56.9%)	0.286 ^c^
Other French regions	49 (37.4%)	27 (41.6%)	
Unknown	0	1 (1.5%)	
French overseas territories ^e^	174 (57.0%)	0	
***C. trachomatis* coinfection—no. (%)**			
Available information	200 (65.6%)	60 (92.3%)	
Coinfection	40 (20.0%)	6 (10.0%)	0.084 ^c^
No coinfection	160 (80.0%)	54 (90.0%)	
Missing information	105 (34.4%)	5 (7.7%)	
**Macrolide treatment anteriority—no. (%)**			
Available information	ND	39 (60.0%)	
Previous macrolide regimen	ND	35 (89.7%)	
No previous macrolide regimen	ND	4 (10.3%)	
Missing information	ND	26 (40.0%)	
**Macrolide resistance—no. (%)**			
Resistant	43 (14.1%)	62 (95.4%)	<0.001 ^d^
Susceptible	262 (85.9%)	3 (4.6%)	

^a^ Men who have sex with men: 56 PrEP users and 9 HIV positive. ^b^ The age average comparison was performed using the Wilcoxon signed-rank test. The distribution of categorical variables was compared by chi-square ^c^ or Fischer’s exact ^d^ tests. ^e^ La Reunion island, French Guyana, and French Polynesia.

**Table 2 microorganisms-10-01587-t002:** Fluoroquinolone resistance-associated mutations in *M. genitalium*-positive patients.

	All Patients	Cohort	
		Women	MSM
***parC* gene**			
Successful amplification	338	276	62
Wild type	291 (86.1%)	251 (90.9%)	40 (64.5%)
Mutations detected ^a^	47 (13.9%)	25 (9.1%)	22 (35.5%)
**Gly81Cys**	**1**	**1**	-
**Ser83Arg**	**1**	**1**	-
Ser83Asn	4	2	2
**Ser83Ile**	**32**	**13**	**19**
**Asp87Asn**	**2**	**2**	-
**Asp87Tyr**	**3**	**3**	-
Ser95Asn	1	1	-
Ile105Phe	1	1	-
His106Tyr	1	1	-
Ala119Val	1	-	1
No amplification	32	29	3
Total of patients	370	305	65
***gyrA* gene**			
Successful amplification	283	227	56
Wild type	279 (98.6%)	225 (99.1%)	54 (96.4%)
Mutations detected	4 (1.4%)	2 (0.9%)	2 (3.6%)
Gly93Cys ^b^	1	-	1
Met95Ile ^b^	1	-	1
Ala105Thr	1	1	-
Asp107Asn	1	1	-
No amplification	55	49	6
Total of patients	338	276	62

^a^*M. genitalium* numbering. ^b^
*gyrA* mutations associated with the Ser83Ile *parC* mutation. Mutations likely of clinical significance are noted in bold. *gyrA* PCR was performed only on samples with successful *parC* amplification.

**Table 3 microorganisms-10-01587-t003:** Epidemiological characteristics of patients and specimens in the three clusters based on *mgpB* typing.

	Cluster A	Cluster B	Cluster C	*p* _A-B_ ^a^	*p* _A-C_ ^b^	*p* _B-C_ ^c^
**Total number of patients** **(*n* = 370)**	**149**	**80**	**141**			
**Cohorts**						
MSM	17 (11.4%)	9 (11.3%)	39 (27.7%)	0.971	<0.001	0.004
Women	132 (88.6%)	71 (88.7%)	102 (72.3%)			
**Geographical origin**						
Metropolitan France	67 (45.0%)	51 (63.8%)	78 (55.3%)	0.007	0.078	0.222
Overseas France	82 (55.0%)	29 (36.2%)	63 (44.7%)			
**Total number of specimens** **(*n* = 374)**	**150**	**81**	**143**			
**Specimen type**						
Vaginal swab	115 (76.7%)	61 (76.3%)	83 (58.0%)	0.400	<0.001	0.021
Urine	28 (18.7%)	12 (15.0%)	33 (23.1%)			
Rectal swab	7 (4.6%)	7 (8.7%)	27 (18.9%)			
**Macrolide resistance**						
Resistant	30 (20.0%)	24 (29.6%)	54 (37.8%)	0.099	<0.001	0.220
Susceptible	120 (80.0%)	57 (70.4%)	89 (62.2%)			
**Fluoroquinolone resistance**						
Resistant	17 (12.5%)	13 (17.8%)	11 (8.3%)	0.297	0.256	0.041
Susceptible	119 (87.5%)	60 (82.2%)	122 (91.7%)			

^a^*p* value for χ^2^ tests between Clusters A and B. ^b^
*p* value for χ^2^ tests between Clusters A and C. ^c^
*p* value for χ^2^ tests between Clusters B and C.

## Data Availability

The data presented in this study are openly available in GenBank, accession nos. MZ553845-MZ553914.

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
