# Peer review of "Molecular Typing Reveals Distinct Mycoplasma genitalium Transmission Networks among a Cohort of Men Who Have Sex with Men and a Cohort of Women in France"

_microorganisms, 2022, doi:10.3390/microorganisms10081587_

Round 1

Reviewer 1 Report

The manuscript by Guiraud et al. present data on M. genitalium molecular typing and determination of AMR determinants in two cohorts of women from France (overseas and Metropolitan) and in MSM failing macrolide treatment (line 71). Although the typing data is interesting, the comparisons between men and women seem to be unjustified as the populations are so different.

Major points of concern

11.       Authors compare uncritically samples from a very selected MSM population experiencing macrolide treatment failure, so all comparisons of macrolide resistance are completely irrelevant when the comparator is more or less unselected women participating in a prevalence study. It is surprising that not 100% MSM were macrolide resistant (only 95%) and it is also confusing that table 1 reports previous macrolide in only 90% of the MSM when macrolide treatment failure was the inclusion criterion.

22.       Comparisons between cohorts are only relevant for factors not influenced by the inclusion criteria for the cohort. Several variables are strongly related: e.g. (l. 231) it is not surprising that macrolide resistance is associated with cluster C, when cluster C is dominated by the MSM cohort preselected for macrolide resistance.

33.       The mgpB typing assay should be much more sensitive than presented in the present work. The typing was performed only on samples positive in the SpeeDx assay, which is not extremely sensitive, so missing 40% is very concerning and points towards technical problems in the assay.

44.       The calculation of the Hunter-Gaston discriminatory index is not appropriate here. The index assumes that samples are unrelated which is not the case here. Some patients even contributed two samples.

55.       Comparisons within cohorts should be analysed with multivariable analysis as many variables are related.

Minor points

11.       Comparison of age with Student’s t-test is probably not appropriate as this variable is unlikely to be normally distributed.

22.       L. 142: Why were asymptomatic MSM tested in the first place? Was that part of the PrEP monitoring at the time of study? It is not recommend to screen this population in the current guidelines.

33.       Was any correction for multiple comparisons used?  

Reviewer 2 Report

This is an interesting paper which investigates M. genitalium genotypes in MSM and females from France. The authors describe several novel genotypes and identified the potential clonal spread of a distinct genotype harbouring antimicrobial resistance. Overall, the paper is of high quality, and I just have a few comments/suggestions: 

- In the abstract you mention that there are 374 samples from 305 women and 65 men, but you don't mention that there were repeat samples from the same patient in the abstract. For clarity, I think either mention the total number of samples (to add up to 374) or mention that there are multiple samples from several patients. 

- The text on Figure 1 is very difficult to read, possibly due to poor image quality. Can you please provide a higher resolution copy of the figure with legible text. When I zoom in on the figure in the document, I can't read the text. 

- Comment: Was macrolide resistance in women lower because you were targeting offshore populations, that are possibly lower socioeconomic areas where the consumption of antimicrobials like macrolides might be lower? It is also noteworthy that you don't have previously antimicrobial usage data for the women in this study, and I think this should be pointed out. It is of course not a huge issue, but something that warrants being highlighted/commented upon in the discussion. 

Reviewer 3 Report

This manuscript describes the genetic relationship between M. genitalium strains and their antibiotic resistance profile in a cohort of men having sex with men and a cohort of women using mgpB/MG309 typing. In general, this is a well designed and performed study.

 1.     Please discuss the possible bias in antibiotic availability and prescription practices in metropolitan and overseas France.

2.     Please discuss the importance of human genetic background as concerns susceptibility to M. genitalium infection (women vs men, different races).
